# Effect of Gamma Radiation on Structural and Optical Properties of ZnO and Mg-Doped ZnO Films Paired with Monte Carlo Simulation

Mivolil Duinong [1], Rosfayanti Rasmidi [1,2], Fuei Pien Chee [1,*], Pak Yan Moh [3,4,*], Saafie Salleh [1], Khairul Anuar Mohd Salleh [5] and Sofian Ibrahim [6]

1. Physics with Electronic/Industrial Physics Programme, Faculty of Science and Natural Resources, Universiti Malaysia Sabah, Jalan UMS, Kota Kinabalu 88400, Sabah, Malaysia
2. Faculty of Applied Sciences, Universiti Teknologi MARA Sabah Branch, Kota Kinabalu Campus, Kota Kinabalu 88997, Sabah, Malaysia
3. Industrial Chemistry Programme, Faculty of Science and Natural Resources, Universiti Malaysia Sabah, Kota Kinabalu 88400, Sabah, Malaysia
4. Water Research Unit, Faculty of Science and Natural Resources, Universiti Malaysia Sabah, Kota Kinabalu 88400, Sabah, Malaysia
5. Malaysia Nuclear Agency for Non-Destructive Testing (NDT), Kajang 43000, Bangi, Malaysia
6. Malaysia Nuclear Agency, MINTec-SINAGAMA, Kajang 43000, Bangi, Malaysia
* Correspondence: fpchee06@ums.edu.my (F.P.C.); pymoh@ums.edu.my (P.Y.M.)

**Abstract:** In space, geostationary electronics located within the outer van Allen radiation belt are vulnerable to gamma radiation exposure. In terms of application, implementing an electronic system in a high radiation environment is impossible via conventional engineering materials such as metal alloys as they are prone to radiation damage. Exposure to such radiation causes degradation and structural defects within the semiconductor component, significantly changing their overall density. The changes in the density will then cause electronic failure, known as the single event phenomena. Thus, the radiation response of material must be thoroughly investigated before the material is applied in a harsh radiation environment, specifically for flexible space borne electronic application. In this work, potential candidates for space-borne application devices: zinc oxide (ZnO) and Mg-doped ZnO thin film with a film thickness of 300 nm, were deposited onto an indium tin oxide (ITO) substrate via radio frequency (RF) sputtering method. The fabricated films were then irradiated by Co-60 gamma ray at a dose rate of 2 kGy/hr. The total ionizing dose (TID) effect of ZnO and Mg-doped ZnO thin films were then studied. From the results obtained, degradation towards the surface morphology, optical properties, and lattice parameters caused by increasing TID, ranging from 10 kGy–300 kGy, were evaluated. The alteration can be observed on the morphological changes due to the change in the roughness root mean square (RMS) with TID, while structural changes show increased strain and decreased crystallite size. For the optical properties, band gap tends to decrease with increased dose in response to colour centre (Farbe centre) effects resulting in a decrease in transmittance spectra of the fabricated films.

**Keywords:** magnesium zinc oxide; zinc oxide; gamma-ray; radiation damage; space application

## 1. Introduction

Over the past years, various radiation effects on widely used semiconductor devices such as conventional silicon- and germanium-based devices have been investigated thoroughly [1–3]. However, emerging research interest in oxide-based semiconductor is rising due to its radiation superiority threshold and its natural radiation properties. At present, the use of metal oxide-based semiconductors is slowly gaining the attention due to its unique potential in replacing conventional silicon-based semiconductor for high radiation environment application [4].

Among metal oxide-based semiconductor, ZnO based semiconductor shows promising result due to its broad application. Belonging in the II-VI semiconductor group, ZnO is known for its direct, wide bandgap ranging within 3.1 to 3.37 eV, has high excitation binding energy of 60 meV, with a hexagonal wurtzite crystal structure [4,5]. Outside the Earth's exosphere, flexible space borne electronics are vulnerable to various types of cosmic radiation. Among the subsets of cosmic radiation, gamma radiation proves to be the most problematic due to its high energy and deeply penetrating electromagnetic radiation attributed to its short wavelength. As gamma ray interacts with materials, generation of defects is inevitable due to its ionizing properties, which in turn causes the ejections of electrons. In addition, atomic displacement, which in turn induces primary knock-on atoms (PKA) due to irradiation, causes atomic lattice defects [6]. Based on past studies, the displacement energies for oxygen and metal of ZnO are both 57 eV. When ZnO based semiconductor is exposed to gamma radiation, a degradation in its electrical parameters seems apparent with a drop in the turn-on voltage attributed to the structural defects present within the semiconductor, which in turn leads to an increase in its ideality factor [7–9]. Furthermore, the changes can also be associated with the combined effects of interface states creation and electron–hole pair generation in the insulating layer.

However, despite the apparent damage caused by radiation exposure, a number of past studies concluded that metal oxide-based semiconductors show greater radiation threshold than conventional semiconductors, thus highlighting the importance and necessity in replacing current conventional semiconductors [5]. Recently, the effect of gamma radiation on the electrical performance of a heterojunction pairing of $ZnO/CuGaO_2$ [7] has been studied and shows varying electrical performance with increasing irradiation while still retaining its semiconductor properties and functionality, despite high radiation exposure [7]. However, the effects of gamma radiation exposure towards Mg-doped ZnO have not been thoroughly investigated yet, specifically under high radiation exposure [4]. The selection of Mg (Group II element) as a dopant has considerable benefits as Mg ion has a similar radius to Zn ion, which in turn allows facilitation of a wider optical bandgap shifting Fermi level through the creation of impurity states that varies with doping concentration [10] where such addition is beneficial in order to improve or modify the electrical and optical properties [11]. Furthermore, the replacement of Zn by Mg should not cause a significant change in the lattice constant and thus preventing a lattice mismatch, making it a suitable pairing for a heterojunction combination as large crystal structure difference between wurzite hexagonal ZnO can cause unstable phase mixing [12].

Among various deposition methods, the radio frequency (RF) sputtering method of ZnO thin film fabrication is commonly used, due to its high precision capabilities in producing a high-quality film, with higher purity, more controlled composition, and greater adhesive thin film strength, resulting a more homogenous fabrication [13]. On the dopant concentration, a small amount of Mg dopant is sufficient, preferably within 3% to 4% [14,15] as past studies reported that the increase in Mg dopant concentration leads to the production of microstrain and changes the transmittance and as well as the near band emission [15].

In this paper, ZnO and Mg-doped ZnO (3% Mg) thin films of 300 nm were fabricated using an RF sputtering method and exposed to [60]Co gamma radiation with a range from 10–300 kGy range. The doses are selected according to the NASA report from Poivey where a study reveals that the highest fifteen years of total dose level received by the part at geostationary orbit is 24 krad (240 Gy) [16]. Therefore, a total dose of 300 kGy is more than adequate to cater to the harsh conditions in the proposed space environment, showcasing its ability to be used for space exploration beyond the boundary of geostationary orbit and possibility for long-term planetary exploration.

The effects of total ionizing dose (TID) on the structural, optical, and morphological properties were characterized for both pre and post irradiation changes. The characterization parameter includes the structural, optical, and morphological properties of both pre- and post-irradiation changes.

## 2. Experimental Methodology

### 2.1. Sample Preparation

The initial stage of this research focuses on the sample preparation. Ahead of fabrication, the substrates were cleaned using 4 different chemicals; decon-90, distilled water, isopropyl alcohol (IPA), and acetone [7]. The cleaning was started with the use of Decon-90, which is an active cleaning agent, while distilled water was used mainly to remove organic impurities, as well as ions and minerals. Meanwhile, IPA was used to clean oil, bacteria, or grease stains. This was followed by the use of acetone to form a hemiacetal reaction, therefore providing a more thorough cleaning inside an ultrasonic bath [8,13]. Once the substrates were cleaned, the ZnO and MgZnO samples were then deposited via the use of an RF magnetron (Torr international) sputtering technique. The deposition parameter were carried out with a working pressure of 3 mTorr, an argon gas flow of 15 standard cubic centimetres per minute (sccm), and a substrate rotation of about 5 rotations per minute (RPM) to ensure film homogeneity.

### 2.2. Sample Characterization

Prior to radiation exposure, the structural, morphological, and optical properties of fabricated samples were analysed. Once completed, the samples were then irradiated with gamma irradiation with a Cobalt-60 source with a decay energy of 1.25 MeV and a radiation rate of 2 kGy/hr [13] with a TID ranging from 10 kGy–300 kGy. This irradiation was performed at MINTec SINAGAMA facility, Malaysia Nuclear Agency.

The structural analyses of the fabricated samples were integrated into the Scherrer equation, as shown in Equation (1), in order to determine the changes on the crystallite grain size with increasing TID. The average crystallite size of Mg-doped ZnO thin films was estimated using Equation (1) [17–19]:

$$d = \frac{K\lambda}{\beta \cos\theta} \tag{1}$$

where, $K$, $\lambda$, $\theta$, and $\beta$ are the shape factor, X-ray wavelength, diffraction angle, and the full width half maximum (FWHM) of the diffraction peak of the thin film, respectively.

In addition, the lattice strain, (epsilon) which is obtained from crystal imperfections, was also calculated using the tangent formula, as shown in Equation (2) [4]:

$$\varepsilon = \beta/4 \ \tan\theta \tag{2}$$

For the morphological parameters, the surface quality of the thin films was evaluated by extracting the data from the roughness analysis. The root mean square (RMS) roughness, $R_q$ were obtained using the atomic force microscopy (AFM), which employs Equation (3);

$$R_q = \sqrt{\frac{\sum(Z_i)^2}{N}} \tag{3}$$

where $Z_i$ is the current $Z$ value, and $N$ is the number of points of the AFM. While the arithmetic average $R_a$ of the absolute values of the surface height deviations was measured from the mean plane and calculated using Equation (4);

$$R_a = \frac{1}{N}\sum_{j=1}^{N}|Z_j| \tag{4}$$

While determination of its optical band gap adapts the utilization of Beer Lambert's Law, as in Equation (5) [7,19]:

$$(\alpha h\upsilon)^{1/n} = \beta(h\upsilon - Eg) \tag{5}$$

### 2.3. Transport Range of Ions Matter (TRIM) Simulation

TRIM (Transport of Ions in the Matter) (SRIM-TRIM 2013 Professional) is a software package concerning the Transport of ions matter and his been continuously upgraded since its first used in 1985 which uses a Core and Bond (CAB) approach to simulate high energy atomic collision [20]. This simulation gives the calculation results for the 3D spread of ions, damage of the target, sputtering, ionization, and phonon production. The total displacement atoms in the lattice are a sum of a number of vacancies and replacement collisions [20]. When a recoil atom stops and is not a replacement atom, then it becomes an interstitial. In this simulation, the incident atom $^{60}$Co has an atomic number $Co_{27}$ and assumed with energy of $E = E_{Co\text{-}60}$. It has a collision within the target with Zn and Mg of atomic number $Zn_{30}$ and $Mg_{12}$. The damage will then be compared to the physical experiment on the matters of overall damage. For the initial parameter consideration, the element for ionizing radiation was set to be $^{60}$Co with an energy of 1.25 MeV with a lateral projection of $0^0$. The input elements were arranged and was set to simulate about 5000 ions.

## 3. Results and Discussion

### 3.1. Structural Properties of ZnO and Mg-Doped ZnO

Based on Figure 1, X-ray diffraction (XRD) analysis perform using RIGAKU Smartlab, the structural properties of pre- and post-irradiation of n-ZnO shows that the thin film retains its crystalline properties regardless of exposure towards radiation, based on its broad wave. However, based on the analysed XRD data, it was revealed that FWHM increases with the increase in TID. The FWHM of thin films is dependent on the length of peak width. As the intensity peak becomes narrower, the crystal quality of thin films is enhanced, leading to a greater crystallite grain size.

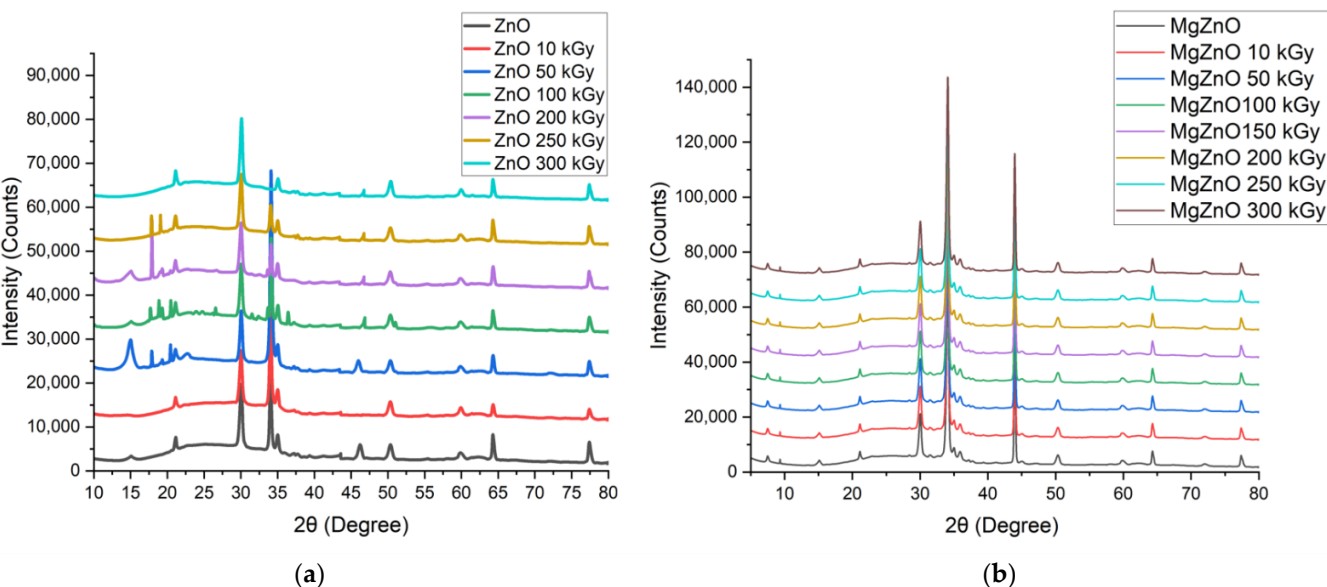

**Figure 1.** *Cont*.

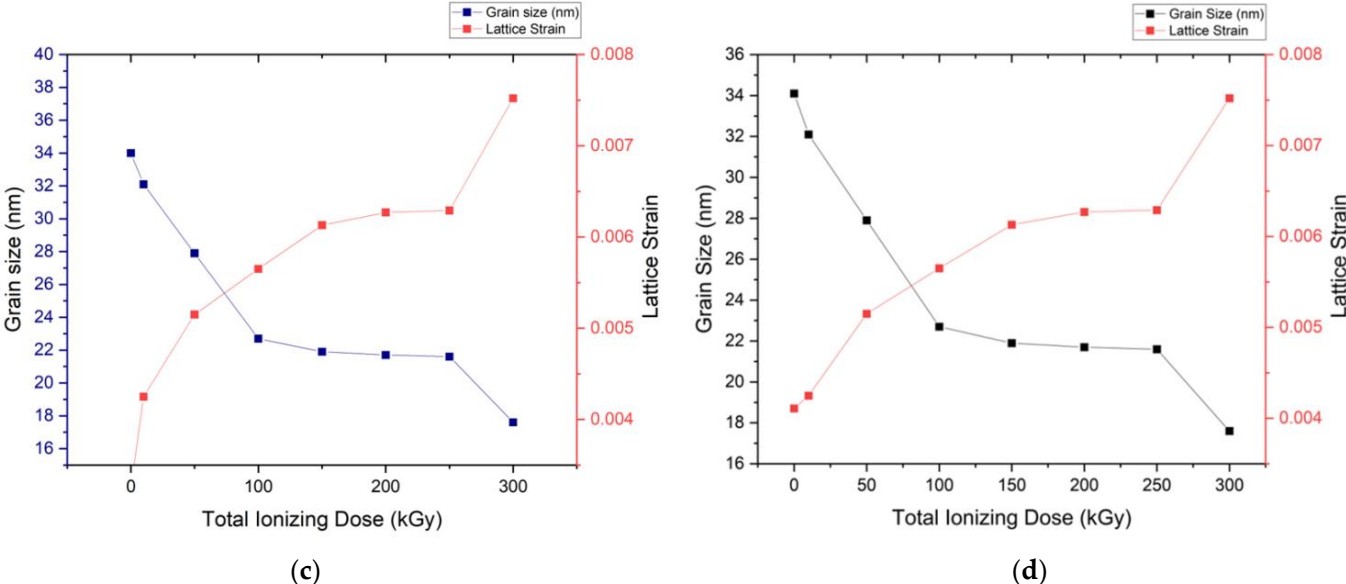

**Figure 1.** XRD of (**a**) ZnO (**b**) MgZnO pre- and post-irradiation changes. The effects of gamma radiation with TID on the lattice parameters of ZnO and MgZnO. (**c**) Grain Size (**d**) lattice strain, respectively.

As ZnO was exposed to gamma ray, the interaction influenced the crystal lattice structure with varying results compared to conventional silicon-based semiconductor [7,11,20]. In general, the production of defects is attributed to the ionization and atomic displacement. During irradiation exposure, the generation of electron–hole pair within the metal oxide-based semiconductor is accumulated around traps [3,7], which triggers generation defects within the crystal lattice [7,20]. Furthermore, the highly energetic gamma interaction with atoms within the lattice site causes lattice displacement, thereby creating a cluster of defects within the crystal structure [20].

The structural lattice defects can be evaluated from the XRD analysis as shown in Figure 1a shows the XRD pattern for different absorbed doses of ZnO thin films deposited on ITO substrates with a specification of sheet resistance of 15 $\Omega$/sq and visible light transmittance of $\geq$85%. The XRD analysis utilizes the technique in analysing the lattice structure, FWHM, crystallite grain size, lattice constants, and lattice strain of the fabricated thin film. The XRD data were extrapolated from 25° to 35° at two positions. By comparing with the ICDD datasheet, it was observed that three distinct XRD spectra peaks for each sample of around 30.07°, 34.007°, and 34.98°, with a peak orientation at (100), (002), and (101) planes, respectively, are present in reference to the (JCPDS card No. 21-1486).

Based on the broadening curve, data suggest a polycrystalline nature of the fabricated samples [4,21]. Analysis on the FWHM values were taken from the highest peak (002). From the diffraction peak selected, various lattice parameters were then calculated utilizing Equations (1) and (2).

The relationship between the lattice strain and crystallite grain size in response to TID is shown in Figure 1c. The XRD analysis revealed that the crystallite grain size diminished with increasing TID. As the relationship of FWHM and crystallite grain size are inverse to one another, this suggests the widening of FWHM with increasing TID. At the highest ionizing dose of 300 kGy, the irradiated ZnO sample shows a lattice strain of 0.00752 with a grain size of 17.60 nm, while unirradiated sample shows a lattice strain of 0.00411 with a grain size of 34.01 nm. This can be attributed to the strong effect of gamma irradiation on the structure of n-ZnO. Moreover, an increased in the lattice strains of MgZnO thin films is observed, in response to the increment of TID. In response, crystallite size was decreased with the increase in gamma absorbed dose, as relationship shows an inverse proportionality between crystallite grain size and lattice strain [4,14] as shown in Figure 1b.

The relationship between changes in the lattice strain and the grain size with increasing TID can be observed in Figure 1d. This result agrees with the research conducted by [4,21], on the changes in metal oxide semiconductor properties towards radiation exposure, where the overall grain size decreases with TID. As the samples were exposed to gamma rays, the electron production due to Compton interaction sets the primary knock-on atoms (PKA's) in motion. This occurs when the incident radiation gains sufficient energy to subsequently displace atoms within the secondary lattice structure, leading to a displacement cascade, resulting in structural distortion and swelling of the grain size [4,21].

On the structural properties of pre- and post-irradiation of Mg-doped ZnO, its crystalline properties are retained despite radiation exposure, similar to ZnO thin film. However, observable changes can be noticed at TID from 50 kGy to 300 kGy. The widening of the diffraction spectra with a lattice orientation of (101) is observed, indicating distortion of typical face-centred cubic structure (FCC) of the Mg-doped ZnO at a higher radiation dose [4].

Figure 1d shows the relationship between lattice changes and the sample's grain size with increasing TID. The results revealed that FWHM of the samples increases with TID. This indicates that the grain size diminishes as radiation dose was increased (as seen in Figure 1d. At the highest TID of 300 kGy, the sample shows a lattice strain of 0.00355 with a grain size of 17.8 nm, while the un-irradiated sample shows a lattice strain of 0.00611 with a grain size of 25.28 nm. This can be attributed to the strong effect of gamma-irradiation on the structure of Mg-doped ZnO. In addition, analysis on the composition of ZnO films was examined via the means of EDX technique as shown in Figure 2 with pre- and post-irradiated ZnO which shows that the films is composed of Zn and O atoms suggesting the film is stoichiometric despite after being irradiated. The presence of additional composition is due to the additional coating of Platinum (Pt) for EDX to be performed.

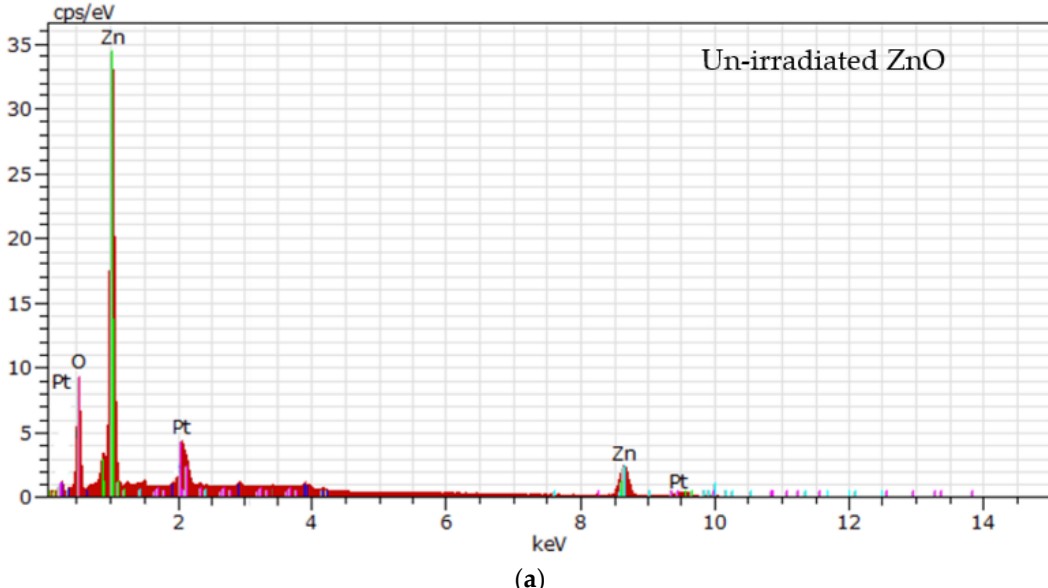

(**a**)

**Figure 2.** *Cont.*

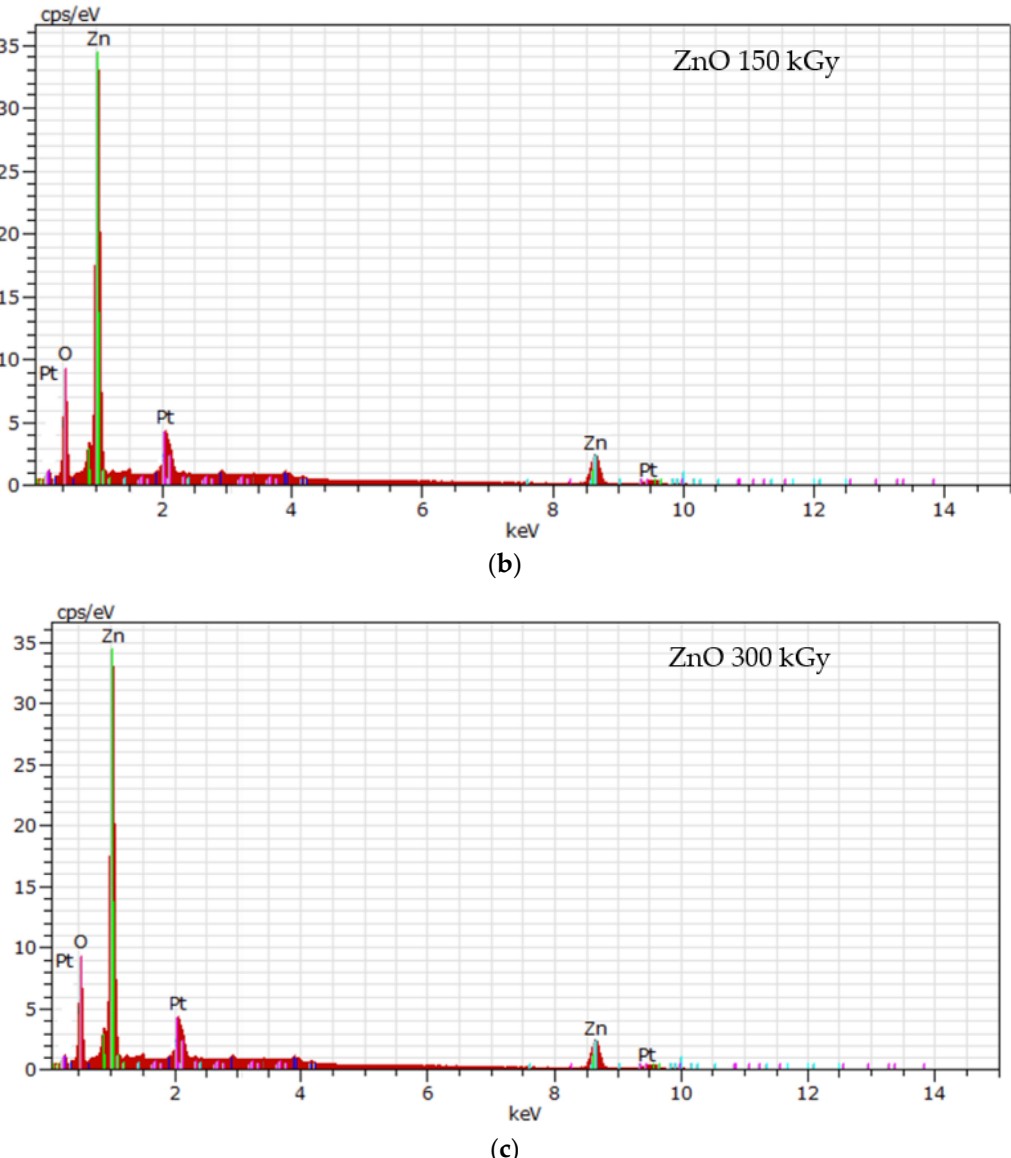

**Figure 2.** EDX of (**a**) ZnO pre- and post-irradiation changes (**b**) 150 kGy and (**c**) 300 kGy.

### 3.2. Optical Properties of ZnO and MgZnO

Optical transmittance (Figure 3a) in the 400–1100 nm wavelength range was investigated using Agilent Cary 60 UV-VIs spectroscopy. As the TID was increased, a significant decrease on transmittance properties was observed. All the irradiated samples show a low transmittance below wavelength of 400 nm. In comparison to the unirradiated sample, varying transmittance levels in the range of 40%–80% above 400 nm were observed. In contrast, the transmittance of other samples (except unirradiated samples) gradually increased above 400 nm. The drop in the transmittance suggests the formation of colour centres. Oxygen vacancies are known as colour centres, or F centres (from Farbe, the German word for colour) [3].

It is believed that ionizing radiation causes structural defects, leading to a change in their density upon exposure to gamma rays. The formation of colour centres has been associated with an increase in electrical conductivity. Free electrons are produced due to band-to-band transitions and trapping of these electrons in oxygen ion vacancies.

Referring to Figure 3a, a sharp cut-off around the 400 nm wavelength in transmittance indicates a direct bandgap material. Therefore, the value of n in reference to Equation (5)

(direct or indirect allowed transition) was taken as 2 to plot the Tauc graph in determining the optical bandgap as shown in Figure 3c.

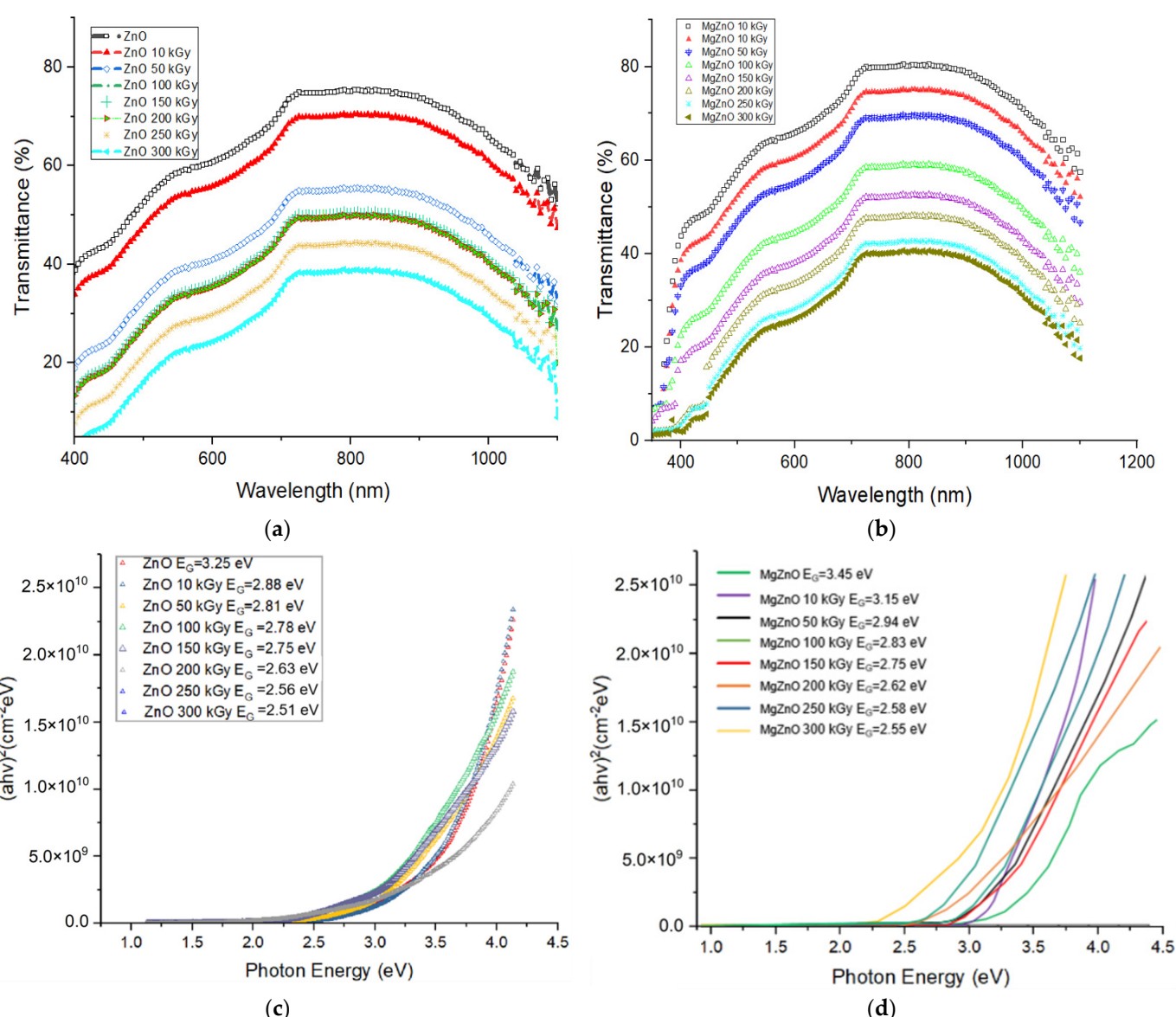

**Figure 3.** Transmittance spectra of (**a**) ZnO and (**b**) MgZnO. The effects of gamma radiation with TID on the band gap parameters of (**c**) ZnO and (**d**) MgZnO.

Based on Figure 4, a blue shift for both ZnO and MgZnO was observed with a decrease in crystallite size with increasing TID. From the obtained analysis, the wavelength of unirradiated ZnO shows a wavelength of 372 nm while at a maximum TID (300 kGy) 366 nm. Separately a shift from 368 nm to 355 nm for MgZnO is also obtained with increasing TID. The blue shift could be related to the decrease in size and quantum confinement effects due to blockage of low energy transition [22] and changes or defects in the morphological structure [23]. The defects are apparent from the XRD data obtained with the broadening of FWHM therefore decreasing the crystallite size. By interpolating the linear portion of the non-linear curve to the x-axis Tauc plot, optical bandgap of the samples at different TID of unirradiated, 10 kGy, 50 kGy, 100 kGy, 150 kGy, and 200 kGy, 250 kGy and 300 kGy were estimated to be 3.25 eV, 2.88 eV, 2.81 eV, 2.78 eV, 2.75 eV, 2.63 eV, 2.56 eV, and 2.51 eV, respectively. The decrease in optical band gap is basically due to the increase in the energy widt of band tails of localized state where similar observations have been reported by past

research [24]. However, consideration of Tauc plot can only be seen at its best a rough estimation and not a true calculation as data extracted from the tangent line were obtained from the graph and can be drawn in several ways [25]. Such considerations were implemented when extracting the point that yields the band gap. Additionally, the equation of a straight line utilizing y = mx + c was used to obtain the linear portion of the non-linear curve to the x-axis of tauc plot.

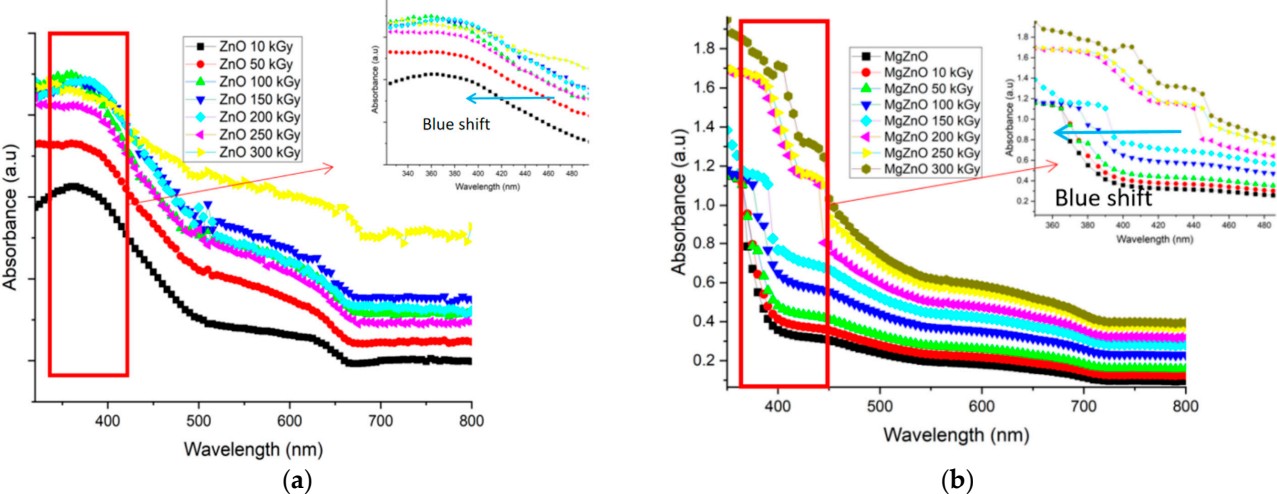

**Figure 4.** Absorbance Spectra with increasing TID of (**a**) ZnO and (**b**) MgZnO.

In ZnO semiconductor, bandgap formation occurs by hybridizing a localized 3d electron from a zinc atom with a 2p electron from an oxygen atom [5]. Figure 3c,d show that bandgap decreases with the increase in TID. Furthermore, an increase in lattice strain and the decrease in grain size crystallites with increasing TID suggest defects strongly associated with increased gamma radiation dose. Thus, the decrease in the band gap is closely associated to the structural defects observed in the structural analysis obtained in the XRD section, with the presence of cluster, cavities, surface roughness, etc. The decrease can also be attributed to the change in the Fermi level between the conduction and valence band [21].

Structural defects due to radiation effects also influence the polarization and spontaneus polarization along the c-axis and a-axis of the local electric fields. This in turn will then lead to band bending at the crystallite boundaries, affecting energy band gap [26]. The Burstein–Moss shift, together with build-in spontaneous polarization and bandgap narrowing phenomenon, determines the overall bandgap [27].

For Mg-doped ZnO, the optical transmittance, ranging from 400 (x-axis) to 1100 nm, shows a similar pattern to that of ZnO (Figure 1b). However, the optical transmittance of Mg-doped ZnO shows a transmittance drop ranging from 85% to 40%. The gradual decrease is present due to the F centre defects. This in turn, however, suggests the increase in absorption, which subsequently supports that Mg-doped ZnO has potential use in UV radiation shielding and UV detection [28,29].

It can be seen clearly that the fabricated Mg-doped ZnO has a larger band gap compared to ZnO. The doping of Mg atom widens the bandgap by increasing the Fermi level within a certain ratio [14]. Despite the increase in the band gap, increasing TID shows a decrement in the bandgap ranging from 10 to 300 kGy, similar to ZnO. It can be observed that exposure to ionizing radiation retains its semiconductor properties falling in the II-IV semiconductor group. The decrease in bandgap is also attributed to the lattice defects incurred upon radiation exposure, where defects can be seen within the structural and morphological changes in Figure 5.

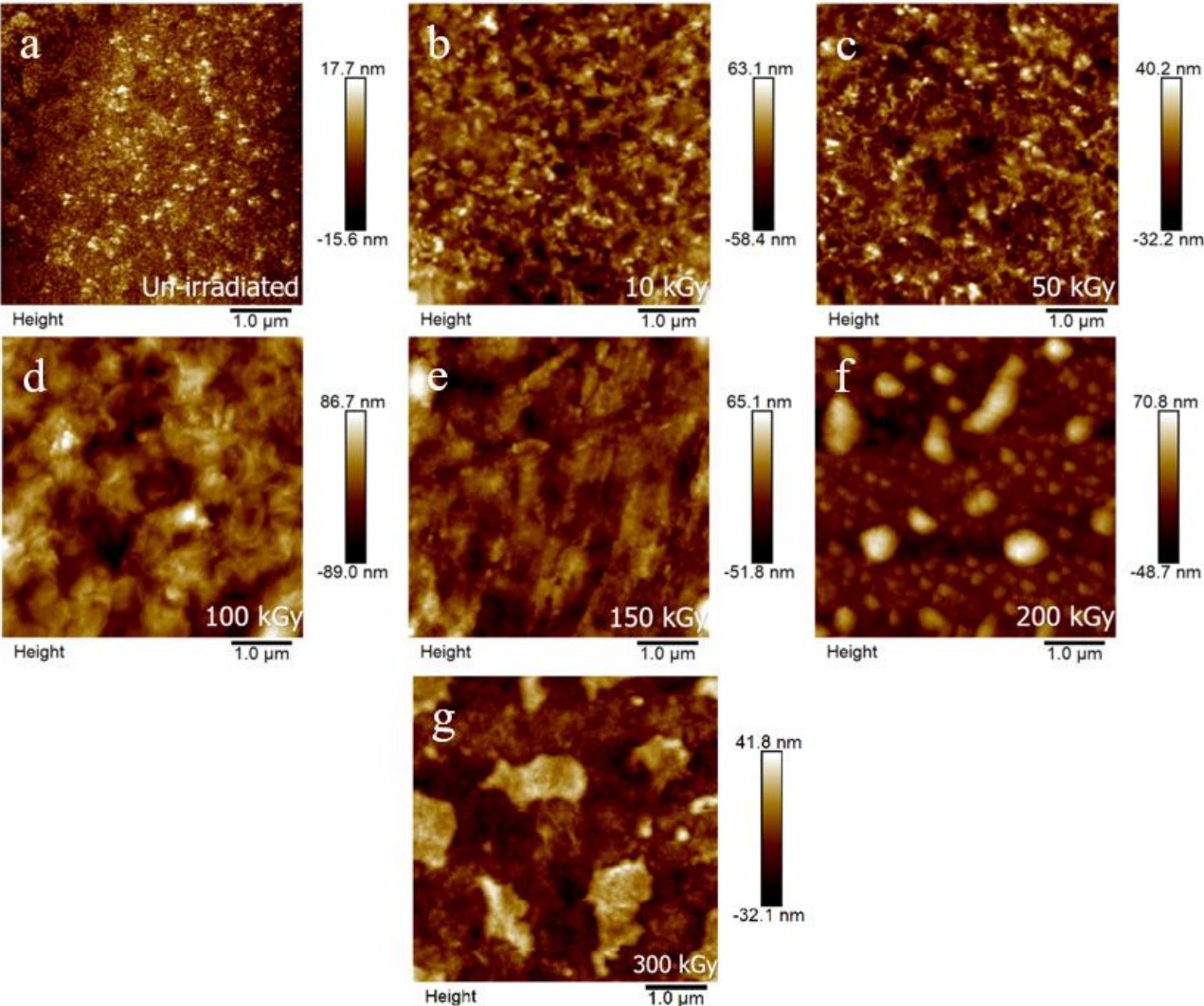

**Figure 5.** AFM images of ZnO with increasing TID; (**a**) Unirradiated, (**b**) 10 kGy, (**c**) 50 kGy, (**d**) 100 kGy, (**e**) 150 kGy, (**f**) 200 kGy, (**g**) 300 kGy.

### 3.3. Morphological Properties of ZnO

The images presented in Figure 4 are obtained over 5 μm × 5 μm scanning range through Bruker Multimode 8 AFM. AFM images and further microstructural analysis reveal good quality homogeneous films when the samples are grown by RF sputtering at 300 °C. The surface roughness of the ZnO thin film was calculated in terms of root mean square (rms) using an AFM software with the average value obtained from 5 spots of each sample. The rms was found to increase monotonously with a mean average from 4.88 nm (un-irradiated) to 82.19 nm (300 kGy). However, a disordered grain and uneven distribution of mixed phase material were found for films exposed to gamma radiation. Both phase and topography images indicate increased surface roughness with TID.

Additionally, the degradation with increasing TID can also be indicated through particle analysis, where the average density of ZnO decreases with a higher dose of radiation where the density count drops from 32.457 ($/\mu m^2$) to 2.532 ($/\mu m^2$). Table 1 shows the analysed parameter of ZnO before and after radiation exposure. Based on past studies, it was revealed that the exposure of radiation caused a significant decrease in the overall grain height and surface mean roughness [30], indicating a similarity about the results of crystallite sizes between the Scherrer's equation results and AFM images which agree with the current findings in this paper.

**Table 1.** Microstructural analysis of ZnO Roughness and particle density.

| Parameter | Unirradiated ZnO | 10 kGy | 50 kGy | 100 kGy | 200 kGy | 250 kGy | 300 kGy |
|---|---|---|---|---|---|---|---|
| Rq (nm) | 4.88 nm | 6.38 nm | 10.88 nm | 22.94 nm | 46.43 nm | 59.39 nm | 82.9 nm |
| Ra (nm) | 3.78 nm | 4.78 nm | 8.32 nm | 18.65 nm | 38.75 nm | 47.51 nm | 78.46 nm |
| Rmax (nm) | 47.0 nm | 57.4 nm | 65.3 nm | 87.12 nm | 101.74 nm | 121.89 nm | 133.52 nm |
| Density ($/\mu m^2$) | 32.457 | 23.715 | 15.322 | 11.0421 | 7.461 | 5.926 | 2.532 |

Rq = RMS surface roughness. Ra = RMS average surface roughness. Rmax = RMS maximum surface roughness.

### 3.4. Morphological Properties of MgZnO

Figure 6 shows the AFM images of the hexagonal nanostructure of MgZnO deposited on ITO substrates with differing TID. From the data analysed, it was observed that increasing TID significantly affects the surface morphology of MgZnO thin film. The increase in TID causes exfoliation and blistering in the hexagonal nanostructure. Notably, the degradation of the morphological surface relates to the non-homogeneity caused by lattice defects with an increase in Rq, from 4.78 nm (un-irradiated) to 45.22 nm (300 kGy). Similar to ZnO, the roughness of said thin films decreases with ionizing dose with a lower particle density count, from 44.32 nm to 1.679 nm. Table 2 shows the analysed parameter of MgZnO pre- and post-irradiation.

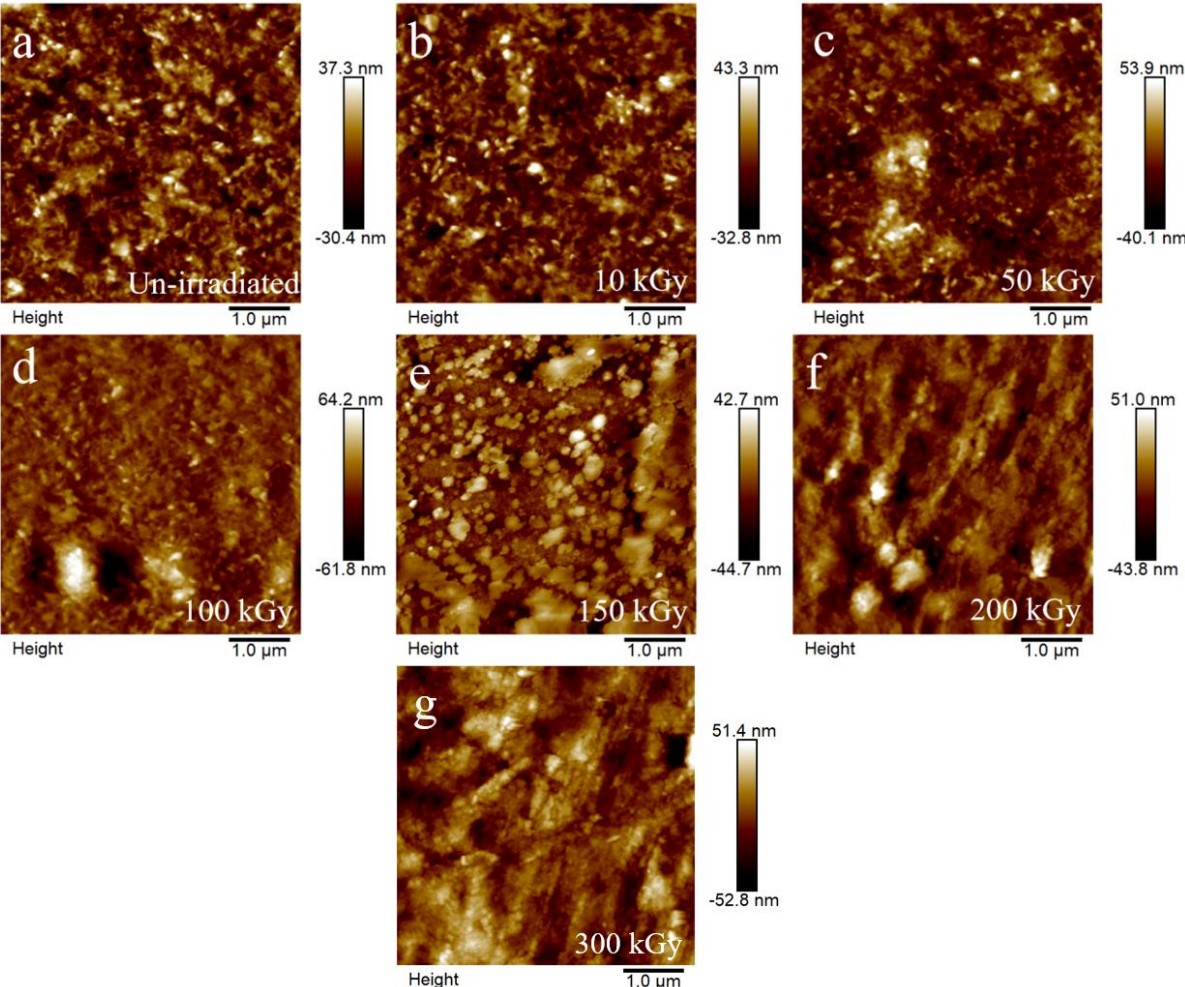

**Figure 6.** AFM images of MgZnO with increasing TID; (**a**) Unirradiated, (**b**) 10 kGy, (**c**) 50 kGy, (**d**) 100 kGy, (**e**) 150 kGy, (**f**) 200 kGy, (**g**) 300 kGy.

**Table 2.** Microstructural analysis of MgZnO roughness and particle density.

| Parameter | Unirradiated ZnO | 10 kGy | 50 kGy | 100 kGy | 200 kGy | 250 kGy | 300 kGy |
|---|---|---|---|---|---|---|---|
| Rq (nm) | 4.78 nm | 7.12 nm | 13.73 nm | 18.82 nm | 22.64 nm | 32.15 nm | 45.22 nm |
| Ra (nm) | 2.89 nm | 4.53 nm | 9.15 nm | 12.11 nm | 15.61 nm | 26.71 nm | 38.11 nm |
| Rmax (nm) | 53.2 nm | 56.0 nm | 64.3 nm | 100.9 nm | 109.2 nm | 142.4 nm | 158.3 nm |
| Density ($/\mu m^2$) | 44.320 | 29.612 | 11.322 | 7.962 | 3.171 | 2.015 | 1.679 |

Rq = RMS surface roughness. Ra = RMS average surface roughness. Rmax = RMS maximum surface roughness.

### 3.5. TRIM-Monte Carlo Simulation

From the simulated result obtained in Figure 7 the ion distribution plots of $^{60}$Co at $E_{Co} = 1.25$ MeV are mostly dominated due to recoiling atoms (blue colour line) for ZnO and (purple colour line) for MgZnO while the other colours' dispersion of dots and clusters, however, indicates the displacement of the targeted atom.

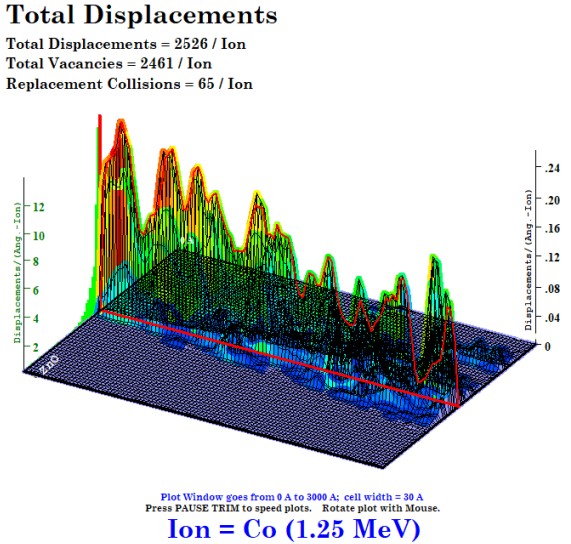

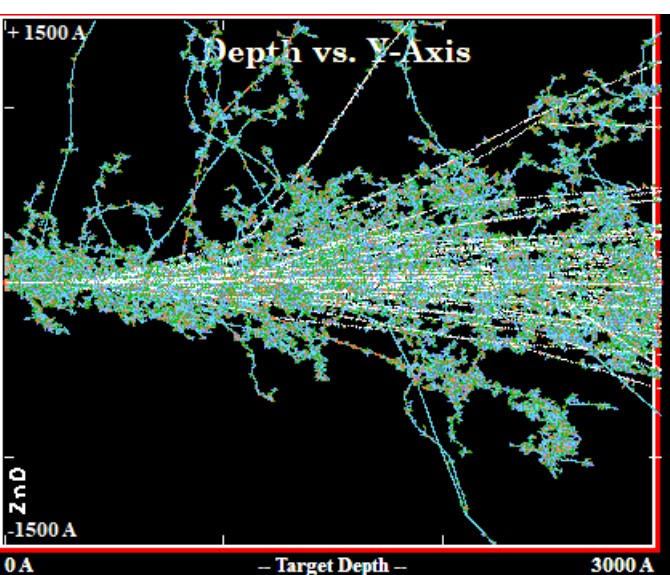

**Figure 7.** 3D plot of the total displacements and energy dispersion within the ZnO cell with the 1.25 MeV radiation.

The simulation is performed under the same parametric thickness and element composition of ZnO with an assumed energy of cobalt-60 at 1.25 MeV as shown in Figure 7. Through the exposure of an incidence energy at 1250 KeV, energy deposition occurs mainly at the surface of the target, displacing a large amount of the lattice atoms from the surface thus, inducing a significant number of recoiling atoms. The recoiling of the other atoms within the target will result in vacancy formation. In addition, recoiling atoms tend to be more active where it has sufficient energy to experience ionization. From the simulated data, the exposure of ZnO films towards cobalt-60 exposure shows to have a significant impact towards the film. This can also be justified where the overall structural and optical properties are affected due to the damage displacement as shown from the simulated result. The damage events are as show in Table 3.

**Table 3.** The damage events and energy absorbed due to Cobalt-60 radiation source at 1250 KeV towards ZnO and MgZnO respectively.

| Incidence Energy (KeV) 1250 | Damage Events | | | | | Energy Absorbed (KeV/Ion) |
|---|---|---|---|---|---|---|
| | Total Vacancies | Replacement Collisions | Target displacement | Ionization KeV/Ion | Target damage KeV/Ion | |
| 1250 | 2461 | 65 | 2526 | 864.6 | 24.46 | 871.06 |
| 1250 | 1758 | 68 | 2700 | 951.7 | 19.22 | 970.92 |

Referring to Figure 8, the simulation of MgZnO, the overall total displacement is lower compared to that of ZnO; however, the addition of Mg dopant to ZnO shows a more significant ionization which suggests that if applied for electrical applications, the vulnerability of MgZnO is more susceptible due to ionizing radiation, which leads to dysfunctionality in digital circuits. In addition, the replacement collision is slightly higher in caparison to ZnO which is caused by the energy absorption by the MgZnO layer. From the physical comparison, the damage events are seen to be equal where both ZnO and MgZnO experience in the degradation of its optical and structural properties while simulation shows that the ionization level of MgZnO is much greater than ZnO therefore suggesting a junction device combination could potentially show that ZnO suffers less in terms electrical degradation. The damage events are as shown in Table 3.

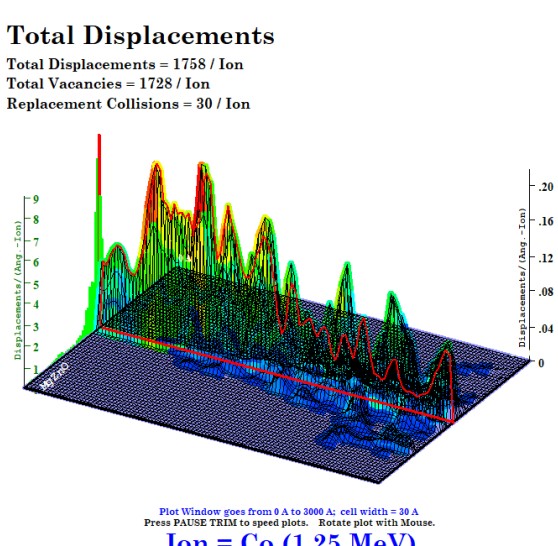
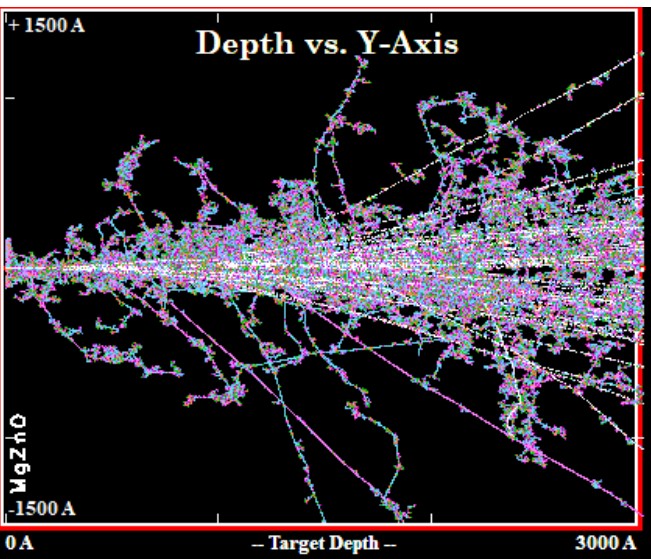

**Figure 8.** 3D plot of the total displacements and energy dispersion within the MgZnO cell with the 1.25 MeV radiation.

## 4. Conclusions

High energy gamma radiation interacts with ZnO and Mg-doped ZnO thin film of thickness 300 nm, deposited via RF sputtering method. The effect of TID of gamma radiation on ZnO and Mg-doped ZnO was studied by incorporating various parameters, which includes the structural, morphological, and optical properties. As the samples were exposed with increased TID, significant changes within the surface morphology and an increase in defects directly influence the transmittance properties of the thin films. Optical bandgap shift is also noticed, which is attributed to induced defects and change in lattice strain. Additionally, both ZnO and Mg-doped ZnO show a similar trend in terms of degradation as analysis on the microstructural grain shows an increased surface roughness and lower density count. In terms of ionization damage, the simulation shows that energy

absorbed by MgZnO is more significant and could potentially be lower in electrical effectiveness compared to ZnO. However, despite the apparent defects, ZnO and Mg-doped ZnO show promising aspects for the application for a high radiation environment where at the maximum dose of 300 kGy it retains within the II-IV semiconductor group properties.

**Author Contributions:** Conceptualization, M.D. and F.P.C.; methodology, M.D. and R.R.; software, M.D and F.P.C.; validation, F.P.C. and P.Y.M.; formal analysis, M.D., R.R., F.P.C. and S.S.; investigation, M.D., R.R., F.P.C. and S.S.; resources, K.A.M.S. and S.I.; data curation, M.D.; writing—original draft preparation, M.D., R.R and F.P.C.; writing—review and editing, M.D., R.R. and F.P.C.; visualization, F.P.C. and P.Y.M.; supervision, F.P.C. and P.Y.M.; project administration, F.P.C., S.S. and P.Y.M.; funding acquisition, F.P.C. and S.S. All authors have read and agreed to the published version of the manuscript.

**Funding:** This research was funded by the Ministry of Higher Education Fundamental Research Grant Scheme (FRGS/1/2020/STG07/UMS/02/1) and Universiti Malaysia Sabah Research Grant Scheme (GUG0482-1/2020).

**Institutional Review Board Statement:** Not applicable.

**Informed Consent Statement:** Not applicable.

**Data Availability Statement:** All data that support the findings of this study are included within the article.

**Conflicts of Interest:** The authors declare no conflict of interest.

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
