# Peer review of "Effect of Gamma Radiation on Structural and Optical Properties of ZnO and Mg-Doped ZnO Films Paired with Monte Carlo Simulation"

_coatings, doi:10.3390/coatings12101590_

Round 1
Reviewer 1 Report
In the paper Effect of Gamma Radiation on Structural and Optical Properties of ZnO and Mg-doped ZnO Films authors investigate the influence of Gamma radiation on properties of ZnO based structures (pure ZnO and Mg doped). Topic of the paper is interested, however the paper have to corrected and revised again.
My main remark is related with doping. Authors used a term of Mg doping and describes a MgZnO film with 3% of Mg. During doping we introduce dopant atoms in small concentration in order of 2x1017 cm-3 for instance. This additional atoms generate free carriers: electrons or holes. Here authors rather alloyed MgO and ZnO and obtained Mg0.03Zn0.97O. Also the discussed properties of MgZnO film are rather related with alloying and therefore satisfy Vegard law.
Secondly, authors claims In general, the production of defects are attributed to the ionization and atomic displacement (lines 166-167) and The formation of colour centres has been associated with an increase in electrical conductivity. Free electrons are produced due to band-to-band transitions and trapping of these electrons in oxygen ion vacancies (lines 241-243) but do not provides any information about basic electrical properties of the analysed films. What about carrier concentration, film resistivity of ZnO and MgZnO layers?
Some additional remarks:
- line 104: The cleaning - interrupted sentence
- lines 104-106 repetition of In contrast
- please provide details about magnetron sputtering system, XRD system, UV-Vis spectrometer (as provided for AFM)
- Figure 1c, 1d - please rotate right Y axis by 180 degree
- Figure 3c - some errors in the figure legend descriptions. Band gap for ZnO irradiated with 200 kGy should be 2.63 instead of 3.63 eV. The same with 250 and 300 kGy samples.
- Tauc instead of tauc (line 253) and Fermi instead of fermi (lines 264 and 277)
- Figure 1c- why values for untreated ZnO are missing?
- line 173 - Figure (a). Figure 1(a) repetition
- In line 260 authors claim that defect created during irradiation affects the properties of deposited films. It is of course true. My question is did authors perform PL measurements and compared the spectra? Irradiation may create different defects which can be reveal by photoluminescence spectra analysis.
- line 268 - authors claim that band gap narrowing is associated with Burstein-Moss effect. It is possible, but B-M effect is connected with change of free carrier concentration while in the paper authors did not mentioned about carrier concentration in analysed films.
- what is a meaning of (1000 nm - 5000 nm) in line 311?
- line 335 - The simulation is simulated sounds little bit funny
- line 338 what K stands for?
Author Response
Please see the attachment for the review outline.

Reviewer 2 Report
Referee report on “Effect of Gamma Radiation on Structural and Optical 2 Properties of ZnO and Mg-doped ZnO Films Paired with Monte Carlo Simulation".
Although this topic is of some interest, this manuscript in its present form cannot be recommended for publication and requires at least some improvement.
1. Line 40-42. This proposal of the most general nature is the first indication of the relevance of the work and should be confirmed by appropriate references. Unfortunately, the first reference [1] of 2017 has only 2 citations (SCOPUS), which does not indicate the above indicated. It is recommended to replace [1] by more appropriate reference. You may consider the following recent review in MDPI journal: Karmakar, A.; Wang, J.; Prinzie, J.; De Smedt, V.; Leroux, P. A Review of Semiconductor Based Ionising Radiation Sensors Used in Harsh Radiation Environments and Their Applications. Radiation 2021, 1, 194-217. https://doi.org/10.3390/radiation1030018
2. Line 57. There is no specific information about the mechanisms of defect formation in ZnO. However, it is well known that the formation of radiation defects in ZnO occurs according to the knock- on mechanism (elastic collisions). The corresponding displacement energies can be found in Table 5 of the review: Popov, A. I., Kotomin, E. A., & Maier, J. (2010). Basic properties of the F-type centers in halides, oxides and perovskites. Nuclear Instruments and Methods in Physics Research Section B: Beam Interactions with Materials and Atoms, 268(19), 3084-3089.
3. Paragraph 2.3. What initial parameters were used for TRIM simulations?
4. Taking into account that the authors reasonably assume the formation of F-type centers, the superposition of the absorption edge with optical absorption bands of the point defects creates serious difficulties in determining the band gap Eg , which is already an ambiguous procedure. . There are many reasons for such claim. A special article was devoted to this issue by the Editors of “Optical Materials” : M.G. Brik, A.M. Srivastava et al , A few common misconceptions in the interpretation of experimental spectroscopic data
Optical Materials, Volume 127, 2022, 112276
https://doi.org/10.1016/j.optmat.2022.112276
Therefore, unambiguous determination of the parameters Eg, the corresponding measurement error requires additional clarification.
In general, the manuscript is interesting and can be recommended for publication after constructive reflection on the above comments.
Author Response

(The authors gave the same response as above.)

Round 2
Reviewer 1 Report
I have found improvement in the revised paper, moreover author responses for my remarks satisfied me. I accept the paper.
Reviewer 2 Report
The authors strongly improved their original manuscript, so it can be now be recommended for publication.